# MSFM: Multi-Scale Fusion Module for Object Detection

## Abstract

Feature fusion is beneficial to object detection tasks in two folds. On one hand, detail and position information can be combined with semantic information when high and low-resolution features from shallow and deep layers are fused. On the other hand, objects can be detected in different scales, which improves the robustness of the framework. In this work, we present a Multi-Scale Fusion Module (MSFM) that extracts both detail and semantical information from a single input but at different scales within the same layer. Specifically, the input of the module will be resized into different scales on which position and semantic information will be processed, and then they will be rescaled back and combined with the module input. The MSFM is lightweight and can be used as a drop-in layer to many existing object detection frameworks. Experiments show that MSFM can bring *+2.5%* mAP improvement with only 2.4M extra parameters on Faster R-CNN with ResNet-50 FPN backbone on COCO Object Detection `minival set`, outperforming that with ResNet-101 FPN backbone without the module which obtains +2.0% mAP with 19.0M extra parameters. The best resulting model achieves a *45.7%* mAP on `test-dev set`. Code will be available.

## 1 Introduction

Object detection is one of the fundamental tasks in computer vision. It requires the detector to localize the objects in the image using bounding boxes and assign the correct category to each of them. In recent years, deep convolutional neural networks (CNNs) have seen great success in object detection, which can be divided into two categories: two-stage detectors, e.g., Faster R-CNN (Ren et al., 2015), and one-stage detectors, e.g., SSD (Liu et al., 2016). Two-stage detectors have high localization and recognition accuracy, while one-stage detectors achieve high inference speed (Jiao et al., 2019). A typical two-stage detector consists of a backbone, a neck, a Region Proposal Network (RPN), and a Region of Interest (ROI) head (Chen et al., 2019). A backbone is a feature extractor usually pre-trained on ImageNet dataset (Deng et al., 2009). A neck could be a Feature Pyramid Network (FPN) (Lin et al., 2017a) that fuses the features from multiple layers. A RPN proposes candidate object bounding boxes, and a ROI head is for box regression and classification (Ren et al., 2015). Compared to two-stage detectors, one-stage detectors propose predicted bounding boxes directly from the input image without the region proposal step, thus being more efficient (Jiao et al., 2019).

One of the key challenges in object detection is to solve the two subtasks, namely localization and classification, coordinately. Localization requires the network to capture the object position accurately, while classification expects the network to extract the semantic information of the objects. Due to the layered structure of the CNNs, detail and position-accurate information resides in shallow but high-resolution layers; however, high-level and semantically strong information exists in deep but low-resolution layers (Long et al., 2014). Another key challenge is scale invariance that the detector is expected to be capable of handling different object scales (Liu et al., 2016).

Feature Fusion is beneficial to object detectors in solving the two challenges. On one hand, through multi-layer fusion (Chen et al., 2020), detail and position information can be combined with semantic information when high and low-resolution features from shallow and deep layers are fused. On the other hand, by fusing the results from different receptive fields (Yu & Koltun, 2016) or scales

(Li et al., 2019) via dilated convolutions or different kernel sizes (Szegedy et al., 2014), objects can be detected in different scales, which improves the robustness of the model.

In this paper, we present a Multi-Scale Fusion Module (MSFM) that extracts both detail and semantical information from a single input but at different scales within the same layer. Specifically, the input of the module will be resized into different scales on which position and semantic information will be processed, and then they will be rescaled back and combined with the module input. The MSFM is lightweight and can be used as a drop-in layer to many existing object detection frameworks, complementing shallow and deep layers with semantic and position information.

Experiments show that MSFM can bring +2.5% mAP improvement with only 2.4M extra parameters on Faster R-CNN with ResNet-50 FPN backbone on COCO Object Detection (Lin et al., 2014) `minival set`, outperforming that with ResNet-101 FPN backbone without the module which obtains +2.0% mAP with 19.0M extra parameters. When applied on other frameworks, it also shows about +2.0% mAP improvement, which show its generalizability. The best resulting model achieves a *45.7%* mAP on `test-dev set`.

## 2 RELATED WORK

### 2.1 MULTI-LAYER FEATURE FUSION

FPN (Lin et al., 2017a) is the de facto multi-layer feature fusion module in modern CNNs to compensate for the position information loss in the deep layer and lack of semantic information in shallow layers. By upsampling the deep features and fusing them with shallow features through a top-down path, it enables the model to coordinate the heterogenous information and enhances the robustness. NAS-FPN (Ghiasi et al., 2019) designs a NAS (Zoph & Le, 2017) search space that covers all possible cross-layer connections, the result of which is a laterally repeatable FPN structure sharing the same dimensions between its input and output. FPG (Chen et al., 2020) proposes a multi-pathway feature pyramid, representing the feature scale-space as a regular grid of parallel bottom-up pathways fused by multi-directional lateral connections. EfficientDet (Tan et al., 2020) adopts a weighted bi-directional feature pyramid network for multi-layer feature fusion. M2Det (Zhao et al., 2018) presents a multi-level feature pyramid network, fusing the features with the same depth and dimension from multiple sequentially connected hourglass-like modules to generate multi-scale feature groups for prediction. Similar structures can also be seen in DSSD (Fu et al., 2017), TDM (Shrivastava et al., 2016), YOLOv3 (Redmon & Farhadi, 2018), and RefineDet (Zhang et al., 2017).

### 2.2 MULTI-BRANCH FEATURE FUSION

In Inception (Szegedy et al., 2014), kernels on Inception Module branches have different sizes, which makes the output of the module contain different receptive fields. However, a large kernel contains a large number of parameters. Instead, dilated convolution allows a kernel to have an enlarged receptive field while keeping the parameter size unchanged. MCA (Yu & Koltun, 2016) utilizes dilated convolutions to systematically aggregate multi-scale contextual information. Going even further, TridentNet (Li et al., 2019) lets multiple convolutions share the same weight but with different dilation rates to explore a uniform representational capability.

## 3 MULTI-SCALE FUSION MODULE

In this section, we present our Multi-Scale Fusion Module (MSFM) and the possible configurations when inserting it into existing frameworks.

### 3.1 MODULE DEFINITION

An instantiation of MSFM is shown in Figure 1a. It can be formulated as follows:

$$M(x) = x + U\{C[F_1(S(x)), F_2(S(x)), ..., F_n(S(x))]\}$$

where $x$ is the module input, $M(x)$ is the module output, $S()$ is the squeeze module that makes the input $x$ thinner, $F_n()$ is the operation on $n$-th branch, $C()$ is the combination function, and $U()$ is the unsqueeze module which will restore the depth of the branch output to make it the same as $x$. The branch operation $F_n()$ can be represented as below:

$$F_n(a) = R_n^{-1}(CGN_{n,i}(CGN_{n,i-1}(...(CGN_{n,1}(R_n(a))))))$$

where $a = S(x)$ is the result of squeeze module, $R_n()$ is the resize function on $n$-th branch, $CGN_{n,i}$ is the $i$-th $\{Conv2D \Rightarrow GroupNormalization \Rightarrow NonLinearity\}$ operation on $n$-th branch, $R_n^{-1}$ is the resize function to restore the feature dimension (height and width).

To make the module lightweight, we utilize a bottleneck-like (He et al., 2015) structure where the module input will first be thinned channel-wise, then fed into the branches. Branch input is resized using bilinear interpolation, and the same method is used when resizing the feature back to its original size. All the 3x3 convolutions on the branches have the padding=1 to keep the spatial dimension unchanged, and the number of the output channel is the same as that of the input channel as well. We choose ReLU as the nonlinearity activation in the MSFM. By default, MSFM is inserted in stages 2, 3, and 4 for ResNet backbones (He et al., 2015).

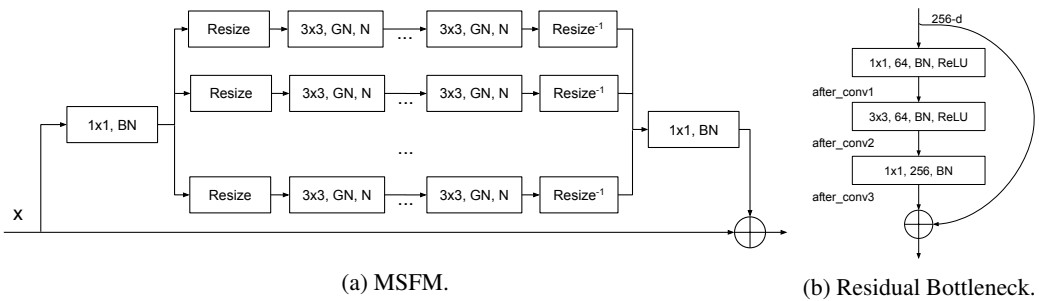

(a) MSFM.

(b) Residual Bottleneck.

Figure 1: MSFM and Residual Bottleneck. BN=Batch Normalization (Ioffe & Szegedy, 2015), N=NonLinearity, GN=Group Normalization (Wu & He, 2018), 1x1=1x1 Convolution, 3x3=3x3 Convolutional with padding=1.

## 3.2 CONFIGURATIONS

MSFM acts as a drop-in layer to existing frameworks. To show several possible configurations when inserting it into an object detector, we take as an example inserting it into a ResNet backbone. A Residual Bottleneck (He et al., 2015) in ResNet (He et al., 2016) is shown in Figure 1b. Some tunable hyperparameters we can configure are listed in Table 1.

Table 1: Tunable hyperparameters

| Name | Description |
| --- | --- |
| Position | Insertion position of the MSFM, after_conv1, after_conv2 or after_conv3 |
| Scales | Scales used to resize the module input on all the branches |
| Ratios | Squeeze ratios used by the first 1x1 Conv to make the module input thinner |
| Norm_group | Number of groups to separate the channels into for Group Normalization |
| Conv_number | Number of {Conv2D, Group Normalization, Nonlinearity} on each branch |
| Fusion_type | Combination method used to fuse the branch results, add or concatenation |

## 4 EXPERIMENTS

To evaluate the proposed module, we carry out experiments on object detection and instance segmentation tasks on COCO (Lin et al., 2014). Experimental results demonstrate that the MSFM can

enhance the performance of common two-stage object detection frameworks with very light computational overhead.

## 4.1 EXPERIMENTS SETUP

We perform hyperparameter tuning on Faster R-CNN with ResNet-50 FPN backbone (Ren et al., 2015). Unless otherwise stated, the backbone of the framework being mentioned is ResNet-50 FPN. To test the generalizability of MSFM, experiments are also conducted on Faster R-CNN with ResNet-101 FPN backbone (Ren et al., 2015), Mask R-CNN (He et al., 2017), Cascade R-CNN (Cai & Vasconcelos, 2017), Grid R-CNN (Lu et al., 2018), Dynamic R-CNN (Zhang et al., 2020), RetinaNet (Lin et al., 2017b), Reppoints (Yang et al., 2019), and Faster R-CNN with ResNet-50 FPN and Deformable Convolution on c3-c5 (Dai et al., 2017). We carry out our experiments on object detection and instance segmentation tasks on COCO (Lin et al., 2014), whose train set contains 118k images, `minival set` 5k images, and `test-dev set` 20k images. Mean average-precision (mAP) scores at different boxes and mask IoUs are adopted as the metrics when evaluating object detection and instance segmentation tasks.

Our experiments are implemented with PyTorch (Paszke et al., 2019) and MMDetection (Chen et al., 2019). The input images are resized such that the shorter side is no longer than 800 pixels. and the longer side is no longer than 1333 pixels. All the models are trained on 8 GPUs with 2 images per GPU. The backbone of all models are pretrained on ImageNet classification dataset (Deng et al., 2009). Unless otherwise stated, all models are trained for 12 epochs using SGD with a weight decay of 0.0001, and a momentum of 0.9. The learning rate is set to 0.02 initially and decays by a factor of 10 at the 8th and 11th epochs. Learning rate linear warmup is adopted for first 500 steps with a warmup ratio of 0.001.

## 4.2 ABLATION STUDIES

The ablation studies are performed on COCO 2017 (Lin et al., 2014) `minival set`.

Unless otherwise stated, the MSFM in the following experiments has the default configuration: the insertion position is after_conv3, the resize scales of three branches are 0.5, 0.7, and 1, respectively, the squeeze ratios are 16, 32, and 64 for stage 2, 3, and 4 of ResNet-50 (He et al., 2015), respectively, the number of groups in Group Normalization (Wu & He, 2018) is 16, only one {Conv2D, Group Normalization, Nonlinearity} operation is adopted on all branches, and the method to combine the branch results is add.

### 4.2.1 SCALES

As can be seen from Table 2 Scales part, small scales (3S=[0.5, 0.7, 1], 5S=[0.5, 0.6, 0.7, 0.85, 1]) are helpful for detecting large objects, while large scales (3L=[1, 1.4, 2]) can enhance the detection of small objects. Compared to only using small or large scales, using compound scales (4=[0.5, 0.7, 1.4, 2], 5=[0.5, 0.7, 1, 1.4, 2]) turn out to be the optimal option, which can achieve better overall performance. This indicates that simultaneously generating and inserting detail and semantic information to the same layer is beneficial.

### 4.2.2 RATIOS

We compare the effect of different squeeze ratios for different insertion positions, shown in Table 2 Ratios part. For position=after_conv3, as we increase the ratios, the model will experience more information loss but less computational overhead; therefore, the ratios of 16, 32, and 64 for stages 2, 3 and 4, respectively, can be a good trade-off between information loss and computational overhead. For postion=after_conv1 (norm_group=8), MSFM is not sensitive to the change of ratios. We guess that it might be because the channel number is already so low after conv1 that changing its channel number will have no further effect.

### 4.2.3 NORM_GROUP

We explore the optimal group number for Group Normalization (Wu & He, 2018) when inserting into different positions. As we can see from the Norm_group part in Table 2, the best group number

Table 2: Ablation Studies

| Name | $AP$ | $AP_{50}$ | $AP_{75}$ | $AP_s$ | $AP_m$ | $AP_l$ | #Param |
|---|---|---|---|---|---|---|---|
| Faster-RCNN R50 FPN | 37.4 | 58.4 | 40.4 | 21.4 | 41.0 | 47.9 | 41.5M |

| Name | Scales | $AP$ | $AP_{50}$ | $AP_{75}$ | $AP_s$ | $AP_m$ | $AP_l$ | #Param |
|---|---|---|---|---|---|---|---|---|
| | 3S | 38.8 | 60.0 | 42.3 | 22.6 | 42.6 | 49.9 | 42.9M |
| | 5S | 38.9 | 60.1 | 42.3 | **23.0** | 42.4 | **50.3** | 43.1M |
| Scales | 3L | 38.6 | 59.8 | 41.8 | **23.0** | 42.6 | 49.5 | 42.8M |
| | 4 | **39** | **60.3** | 42.3 | 22.2 | **42.8** | 49.7 | 43.0M |
| | 5 | 38.9 | 59.9 | **42.4** | 22.7 | 42.7 | **50.3** | 43.1M |

| Name | Ratios | Pos | $AP$ | $AP_{50}$ | $AP_{75}$ | $AP_s$ | $AP_m$ | $AP_l$ | #Param |
|---|---|---|---|---|---|---|---|---|---|
| | 8,16,32 | 3 | **39.1** | **60.1** | **42.6** | **22.8** | **42.8** | **50.4** | 44.9M |
| | 16,32,64 | 3 | 38.8 | 60.0 | 42.3 | 22.6 | 42.6 | 49.9 | 42.9M |
| Ratios | 32,64,128 | 3 | 38.7 | 59.8 | 42.0 | 22.5 | 42.5 | 50.1 | 42.1M |
| | 4,8,16 | 1 | 38.9 | **60.0** | 42.3 | 22.7 | 42.5 | 50.3 | 42.1M |
| | 8,16,32 | 1 | 38.9 | 59.9 | **42.5** | 22.1 | 42.6 | **50.5** | 41.8M |
| | 16,32,64 | 1 | 38.9 | 59.8 | 42.4 | **22.9** | **42.7** | 50.1 | 41.6M |

| Name | #Group | Pos | $AP$ | $AP_{50}$ | $AP_{75}$ | $AP_s$ | $AP_m$ | $AP_l$ | #Param |
|---|---|---|---|---|---|---|---|---|---|
| | 4 | 3 | 38.7 | 59.8 | 42.4 | 22.5 | 42.4 | 49.8 | 42.9M |
| | 8 | 3 | 38.9 | 59.8 | 42.1 | 22.2 | 42.5 | **50.4** | 42.9M |
| | 16 | 3 | 38.8 | 60.0 | 42.3 | 22.6 | 42.6 | 49.9 | 42.9M |
| | 32 | 3 | **39.1** | **60.2** | **42.5** | **23.0** | **42.8** | 50.3 | 42.9M |
| Norm | 1 | 2 | 38.6 | 59.5 | 42.0 | 22.4 | 42.1 | 50.2 | 41.6M |
| group | 4 | 2 | **38.8** | **59.8** | **42.3** | **22.9** | 42.1 | **50.4** | 41.6M |
| | 8 | 2 | 38.7 | **59.8** | 42.0 | 22.5 | **42.6** | 50.0 | 41.6M |
| | 1 | 1 | 38.5 | 59.4 | 42.4 | 22.1 | 42.3 | 49.8 | 41.6M |
| | 4 | 1 | 38.8 | **59.8** | **42.5** | 22.8 | 42.4 | 50.0 | 41.6M |
| | 8 | 1 | **38.9** | 59.8 | 42.4 | **22.9** | **42.7** | **50.1** | 41.6M |

| Name | #Conv | Scales | $AP$ | $AP_{50}$ | $AP_{75}$ | $AP_s$ | $AP_m$ | $AP_l$ | #Param |
|---|---|---|---|---|---|---|---|---|---|
| | 1 | 3S | 39.1 | 60.2 | 42.5 | 23.0 | 42.8 | 50.3 | 42.9M |
| | 2 | 3S | 39.2 | 60.3 | 42.5 | 22.6 | 42.8 | 50.7 | 43.2M |
| | 2* | 5 | 38.9 | 60.2 | 42.2 | 23.3 | 42.7 | 50.3 | 43.3M |
| Conv | 2 | 5 | **39.6** | **60.6** | **43.4** | **23.7** | **43.1** | **51.3** | 43.7M |
| num | 2* | 3L | 38.9 | 60.2 | 42.4 | 23.1 | 42.8 | 50.2 | 43.1M |
| | 2 | 3L | 39.0 | 60.2 | 42.6 | 23.3 | 42.6 | 49.9 | 43.2M |
| | 2* | 4 | 39.2 | 60.3 | 42.6 | 23.2 | 43.0 | 50.6 | 43.2M |
| | 2 | 4 | 39.2 | 60.1 | 42.7 | 22.7 | 42.9 | 50.6 | 43.5M |

| Name | Type | Pos | $AP$ | $AP_{50}$ | $AP_{75}$ | $AP_s$ | $AP_m$ | $AP_l$ | #Param |
|---|---|---|---|---|---|---|---|---|---|
| | add | 3 | 38.8 | 60.0 | 42.3 | 22.6 | 42.6 | 49.9 | 42.9M |
| Fusion | cat | 3 | 39.0 | 60.2 | 42.3 | 22.3 | 42.8 | 50.5 | 43.8M |
| type | add | 1 | 38.9 | 59.8 | 42.4 | 22.9 | 42.7 | 50.1 | 41.6M |
| | cat | 1 | 39.1 | 60.1 | 42.7 | 23.2 | 42.7 | 50.7 | 41.7M |

for after_conv3, after_conv2 and after_conv1 are 32, 4, and 8, respectively. Because the channel number is much larger for after_conv3 compared to after_conv1 and after_conv2, the group number for Group Normalization (Wu & He, 2018) is much larger for after_conv3.

### 4.2.4 CONV_NUM

All the experiments of Conv_num in Table 2 are conducted with Norm_group=32. 2* indicates that only the branches with scales larger than 1 have 2 {Conv2D, Group Normalization, Nonlinearity}

operations. As we can see, the model with scale=[0.5, 0.7, 1, 1.4, 2] and conv_num=2 achieves the best performance. What's more, all the models of conv_num=2 achieves better or at least comparable performance with that of conv_num=2*, which indicates that a coordinate representational power among all the branches is important, even though they do not have the same receptive field size.

### 4.2.5 FUSION_TYPE

As two typical feature fusion operations, add and concatenation are alternatives. We compare their effects in the models of position=after_conv1 and the ones of position=after_conv3. The results in Table 2 show that concatenation is slightly better than add.

### 4.2.6 MULTI-POSITION INSERTION

According to the experiment results and analysis above, we carry out a multi-position insertion ablation study, in order to see the effect of MSFM being inserted in multiple positions. All the experiments in this part have the following configurations for all the models: the resize scales of all the branches are 0.5, 0.7, 1, 1.4, and 2, the squeeze ratios for stage 2, 3, and 4 are 16, 32, and 64, respectively, the number of {Conv2D, Group Normalization, Nonlinearity} operations on all branches is 2, and the combination method is add. The number of groups used in Group Normalization (Wu & He, 2018) is 8, 4, and 32 for after_conv1, after_conv2, and after_conv3, respectively. As can be seen from the results in Table 4, the combination of after_conv2 and after_conv3 turns out the best configuration, which we will use as the default configuration when applying the MSFM to other frameworks.

Table 3: Mutli-position insertion

| Position | $AP$ | $AP_{50}$ | $AP_{75}$ | $AP_s$ | $AP_m$ | $AP_l$ | #Param |
|---|---|---|---|---|---|---|---|
| 1, 2 | 39.3 | 60.2 | 42.8 | 23.1 | 43.0 | 50.9 | 41.8M |
| 1, 3 | 39.3 | 60.2 | 42.8 | 22.9 | 43.2 | 50.7 | 43.9M |
| **2, 3** | **39.9** | **61.0** | **43.5** | **23.5** | **43.7** | **51.6** | 43.9M |
| 1, 2, 3 | 39.3 | 60.4 | 42.6 | 22.8 | 42.9 | 50.6 | 44.0M |

Table 4: Mutli-position insertion for object detection. * indicates with MSFM.

| Framework | $AP$ | $AP_{50}$ | $AP_{75}$ | $AP_s$ | $AP_m$ | $AP_l$ | #Param |
|---|---|---|---|---|---|---|---|
| Faster R-CNN | 37.4 | 58.4 | 40.4 | 21.4 | 41.0 | 47.9 | 41.5M |
| Faster R-CNN* | **39.9** | 61.0 | 43.5 | 23.5 | 43.7 | 51.6 | 43.9M |
| Cascade R-CNN | 40.4 | 58.7 | 44.2 | 22.7 | 43.8 | 53.0 | 69.2M |
| Cascade R-CNN* | **42.6** | 61.5 | 46.6 | 24.9 | 46.3 | 56.3 | 71.5M |
| Grid R-CNN | 39.1 | 57.3 | 42.3 | 22.5 | 43.0 | 50.0 | 64.3M |
| Grid R-CNN* | **41.2** | 60.0 | 44.4 | 24.0 | 45.0 | 52.2 | 66.7M |
| Dynamic R-CNN | 38.9 | 57.5 | 42.5 | 21.4 | 42.5 | 51.4 | 41.5M |
| Dynamic R-CNN* | **40.6** | 59.3 | 44.3 | 23.8 | 43.7 | 53.8 | 43.9M |
| Faster R101 FPN | 39.3 | 60.0 | 42.8 | 22.2 | 43.5 | 51.3 | 60.5M |
| Faster R101 FPN* | **41.2** | 62.1 | 44.9 | 23.6 | 45.5 | 53.9 | 65.8M |
| RetinaNet | 36.4 | 55.3 | 38.8 | 20.7 | 40.0 | 47.1 | 37.7M |
| RetinaNet* | **38.7** | 58.1 | 41.4 | 22.7 | 42.5 | 50.9 | 40.1M |
| Faster R-CNN Dconv | 41.2 | 62.5 | 45.1 | 24.1 | 44.8 | 54.8 | 42.1M |
| Faster R-CNN Dconv* | **42.1** | 63.6 | 46.0 | 25.3 | 45.7 | 54.8 | 44.4M |
| Reppoints | 36.8 | 56.5 | 39.6 | 20.8 | 41.0 | 48.6 | 36.6M |
| Reppoints | 38.5 | 58.7 | 41.5 | 22.4 | 43.2 | 50.4 | 38.9M |
| Mask R-CNN | 38.1 | 58.6 | 41.6 | 21.7 | 41.5 | 49.3 | 44.2M |
| Mask R-CNN* | **40.3** | 61.1 | 43.9 | 23.1 | 44.0 | 52.4 | 46.5M |

Table 5: Mutli-position insertion for instance segmentation. * indicates with MSFM.

| Framework | $AP^m$ | $AP^m_{50}$ | $AP^m_{75}$ | $AP^m_s$ | $AP^m_m$ | $AP^m_l$ | #Param |
|---|---|---|---|---|---|---|---|
| Mask R-CNN | 34.5 | 55.5 | 37.0 | 18.0 | 37.6 | 46.9 | 44.2M |
| Mask R-CNN* | **36.3** | 57.9 | 38.8 | 19.2 | 39.8 | 49.2 | 46.5M |

### 4.3 RESULTING MODELS

To test the generalizability of the proposed MSFM, we apply it to multiple frameworks. The results are shown in Table 4 and Table 5. For a fair comparison, all baseline models are re-trained. As we can see, there is a consistent improvement in the following models when the MSFM is applied, which demonstrates that the MSFM can be used as a drop-in layer for many existing object detection frameworks. Notice that when MSFM is applied to Faster R-CNN with ResNet FPN backbone (Ren et al., 2015), the performance of the model even surpasses the one with ResNet-101 FPN backbone. It indicates that adding the MSFM to existing frameworks is more efficient than just adding more convolutional layers.

We also train a Cascade R-CNN with ResNet-101 FPN backbone for 24 epochs using multi-scale training and submit the results to the evaluation server. The result in Table 6 shows it achieves a *45.7%* mAP on the `test-dev` set.

Table 6: Result of Cascade R-CNN with ResNet-101 FPN backbone trained for 24 epochs with multi-scale training.

| Dataset | $AP^m$ | $AP^m_{50}$ | $AP^m_{75}$ | $AP^m_s$ | $AP^m_m$ | $AP^m_l$ | #Param |
|---|---|---|---|---|---|---|---|
| minival | 45.4 | 64.3 | 49.6 | 27.8 | 49.3 | 58.9 | 93.4M |
| test-dev | 45.7 | 65.0 | 49.8 | 27.4 | 48.7 | 57.1 | |

## 5 CONCLUSION

In this paper, we have presented a Multi-Scale Fusion Module (MSFM) that extracts both detail and semantical information from a single input but at different scales within the same layer. Ablation studies have demonstrated that MSFM can bring *+2.5%* mAP improvement with only 2.4M extra parameters on Faster R-CNN with ResNet-50 FPN backbone on COCO Object Detection `minival set`, outperforming that with ResNet-101 FPN backbone without the module which obtains +2.0% mAP with 19.0M extra parameters. The best resulting model on Cascade R-CNN with ResNet-101 FPN backbone achieved a 45.7% mAP on COCO Object Detection `test-dev` set.

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
