# OpenReview forum: "MSFM: Multi-Scale Fusion Module for Object Detection"
_ICLR.cc/2021/Conference — Reject_

### Official Review · AnonReviewer3 · 2020-10-17
**Lack of novelty is the main problem**

**Rating:** 3
**Confidence:** 4

**Review:**

This paper proposes to obtain multi-scale features by `resize -> convolution -> resize (inverse)’. Extensive experimental results on COCO validate the effectiveness of the proposed approach.

Pros:

Experimental results on the widely used benchmarking dataset.
Comparison with recent approaches.
Paper is easy to understand, simply because the proposed method is simple.

Cons:
Lack of novelty. The use of multi-scale features is not new [a, b]. The formulation in this paper is not very different from that in [b]. It has been used for lots of applications. In object detection, dilated conv in TridentNet is not the only approach using multiple branches. Approaches like [c, d] also used multi-scale-multi-layer features by resizing.

[a] He, Kaiming, Xiangyu Zhang, Shaoqing Ren, and Jian Sun. "Spatial pyramid pooling in deep convolutional networks for visual recognition." IEEE transactions on pattern analysis and machine intelligence 37, no. 9 (2015): 1904-1916.

[b] Yang, Wei, Shuang Li, Wanli Ouyang, Hongsheng Li, and Xiaogang Wang. "Learning feature pyramids for human pose estimation." In proceedings of the IEEE international conference on computer vision, pp. 1281-1290. 2017.

[c] Gidaris, Spyros, and Nikos Komodakis. "Object detection via a multi-region and semantic segmentation-aware cnn model." In Proceedings of the IEEE international conference on computer vision, pp. 1134-1142. 2015.

[d] Zeng, Xingyu, Wanli Ouyang, Junjie Yan, Hongsheng Li, Tong Xiao, Kun Wang, Yu Liu et al. "Crafting gbd-net for object detection." IEEE transactions on pattern analysis and machine intelligence 40, no. 9 (2017): 2109-2123.


The ablation study in the experimental results did not compare with existing works, like TridentNet, and [c, d] to justify why another multi-scale approach is needed.

Symbols are not illustrated well (Authors need not answer this in the rebuttal but need to revise in the revised version).
`S() is the squeeze module: What is the meaning of squeeze module?
There are no common definitions on `makes the input x thinner’, `combination function’, `unsqueeze module’.

---

> ### Author Response · Authors · 2020-11-24
> **Thanks for your advice!**
>
> Thank you very much for your advice. We will keep working on it based on your comments!

---

### Official Review · AnonReviewer2 · 2020-10-19
**Strong results, limited technical contribution & analysis**

**Rating:** 4
**Confidence:** 5

**Review:**

In this paper, the authors study the problem of scale-friendly feature fusion for object detection. Specifically, the authors propose to process features at each layer of a feature pyramid network at multiple scales and fuse them back into a single scale. To be specific, they resize features at a layer into multiple scales, process these rescaled features independently, rescale them back into the original scale and combine them with the original features.

Strengths:
- Scale is an important problem in object detection and the paper addresses an important issue.
- Strong results showing significant improvements, around ~2AP, over baselines, including strong detectors like RepPoints.
- Overall, the paper is very well written. I didn't find any typos or grammatical errors, which is very rare for a thorough reviewer like me.

Weaknesses:
- The novelty is limited. Multi-scale processing at a layer has been extensively studied with ResNext-type and inception-type architectures. The paper just takes such ideas and use them with FPN without any technical or theoretical insights or contributions.

In other words, this appears to be just FPN with inception module. It would have been nicer for the authors to motivate how/whether the novelty goes beyond this.

- The paper does not make a comparison with multi-scale backbone networks such as ResNext. It has been shown that such multi-scale architectures improve the detection performance compared to ResNet type architectures, and this comparison is very crucial for the reader to grasp the significance of the contribution, if any.

- The paper does not make a comparison with respect to methods trying to address limitations of FPN. There are many papers that extend FPN to address scaling issues. It is very crucial for the reader to see how the proposed solution performs in comparison with those methods.

- It would have been nicer to see in Table 4 the details on the type of the backbone used. This might be a very crucial factor in analysing the differences in the gap among the different models.

***AFTER AUTHOR RESPONSE***

I have read the comments of the other reviewers, which revealed that all reviewers identified the same major issues with the paper (novelty and evaluation). The authors did not provide a rebuttal but kindly thanked the reviewers and stated their intention for improving the paper with the reviewer comments and submitting it for a future venue. Therefore, I changed my overall rating to rejection.

---

> ### Author Response · Authors · 2020-11-24
> **Thanks for your advice!**
>
> Thank you very much for your advice. We will keep working on it based on your comments!

---

### Official Review · AnonReviewer4 · 2020-10-29
**Solid work on experiments but need to improve on writing**

**Rating:** 3
**Confidence:** 4

**Review:**

This paper proposes a new general feature fusion operation, Multi-Scale Fusion Module (MSFM). By adding MSFM layers between feature extraction layers, it is observed that the detection result is improved with minor added parameters.

Pros:
Good to see new work to explore various of ways to perform feature fusion.

Cons:
Major comments:
(1) The quality and clarify of the paper needs to be improved, for example, Table 3 has shifted horizontal line.
(2) Ablation studies clarifies on the effects of the change of configurations, but does provide much evidence on why MSFM module helps with the detection task.

Some minor comments:
(1) Figure1a. could be further clarified by adding the notations mentioned in the equations to to the figure.
(2) Good to report the variances/confidence-intervals of the metrics as well.

---

> ### Author Response · Authors · 2020-11-24
> **Thanks for your advice!**
>
> Thank you very much for your advice. We will keep working on it based on your comments!

---

### Official Review · AnonReviewer1 · 2020-11-01
**The multi-scale feature fusion block does not have enough novelty and is computationally expensive**

**Rating:** 3
**Confidence:** 5

**Review:**


The paper proposes a multi-scale feature fusion block and inserts the block into ResNet backbones for object detection. It is very similar to the inception block in IneceptionNets. The only difference is the proposed feature fusion contains feature map upsampling and downsampling (resize and resize^{-1}) for different branches. The paper has some merits as follows.

1. The method has evaluated on different object detection frameworks, such as Faster R-CNN, Cascade R-CNN, Grid R-CNN, Dynamic R-CNN, RepPoints, etc.
2. The method obvious performance gains on different frameworks with a few additional parameters.

However, the flaws are obvious as follows.
1. The novelty is very limited. Inception block is widely used in deep learning. The paper is only a small modification over Inception. The novelty is far below the bar of ICLR.
2. The proposed feature fusion block is very computation expensive. The upsampling procedure makes the computation cost of the 3$\times$3 convolutions very high. The paper only reports additional parameters but does not report the additional computation cost (e.g., additional FLOPs and running time).
3. Related methods are not compared. At least, inserting inception blocks into ResNets can be compared.

---

> ### Author Response · Authors · 2020-11-24
> **Thanks for your advice!**
>
> Thank you very much for your advice. We will keep working on it based on your comments!

---

### Decision · Program_Chairs · 2021-01-07
**Final Decision**

**Decision:**

Reject

**Comment:**

This submission proposes an approach for fusing representations at multiple scales to improve object detection systems. Reviewers thought the paper was well-written and showed positive results on COCO, a common object detection benchmark. However, reviewers agreed that there was not sufficient methodological novelty or empirical improvement over existing approaches to warrant acceptance at ICLR: several prior works have addressed multiscale fusion and reviewers did not find the evaluation/ablations sufficient to demonstrate the approach yielded substantial improvements over these existing approaches. I hope the authors will consider resubmitting the paper after refining it based on the reviewers' feedback.